# Study on the Properties and Mechanism of Magnesium Oxychloride Adhesive Particleboard Modified by Fly Ash

**DOI:** 10.3390/ma15082845

**Published:** 2022-04-13

**Authors:** Meng Cui, Nihua Zheng, Shixin Wang, Bin Luo, Hongguang Liu

**Affiliations:** 1College of Materials Science and Technology, Beijing Forestry University, Beijing 100083, China; cui0206@bjfu.edu.cn (M.C.); zheng_nihua@163.com (N.Z.); hwangshixin@163.com (S.W.); luobincl@bjfu.edu.cn (B.L.); 2Beijing Building Materials, Beijing Building Materials Testing Academy Co., Ltd., Beijing 100024, China

**Keywords:** fly ash, magnesium oxychloride adhesive, particleboard, modification, mechanism

## Abstract

Magnesium oxychloride adhesive (MOA) is a kind of inorganic adhesive with a low energy consumption and environmental protection. The modified particleboard with magnesium oxychloride adhesive (MOPB) has the advantages of no volatiles and has a high mechanical strength. In this study, MOA and poplar shavings were used to prepare MOPB, fly ash (FA) was used to modify and enhance the properties of MOPB; the influence mechanism on the mechanical of MOPB was studied. Fourier transform infrared spectroscopy (FTIR) and X-ray diffraction (XRD) were used to analysis the bonding mechanism of MOA and poplar shavings, and the composite system model of MOPB was constructed. The mechanism of modified MOPB with FA (MOPB-FA) was clarified. The results show that the use of FA effectively improved the mechanical strength of MOPB, and when the ratio of FA addition was 15 wt%, MOPB-FA’s modulus of rupture (MOR) value was 16.32 MPa, an increase of 24.5% more than before (13.11 MPa). The modulus of elasticity (MOE) value was 4595.51 MPa, an increase of 76.7% more than before (2600 MPa), and the thickness of swelling (TS) value was 0.35%, a decrease of 85.2% less than before (2.36%).

## 1. Introduction

Green manufacturing is the only way to go for China wood-based panel industries [1]. Particleboard, such as wood-based panel, can insulate the noise and improve the temperature and humidity because the inner staggered structure has some room and interspace [2]. Nowadays, concerns in the particleboard industry on development are whether it is eco-friendly and low-carbon [3], which means a high-performance eco-friendly adhesive is the first choice. Studies show that magnesium adhesive has potential in particleboard development [4]. Magnesium adhesive has a stable structure and high compressive strength. Too, it is resistant to fire, water, and corrosion, and without formaldehyde element, it is an optimal adhesive [5]. However, magnesium adhesive has a poor tenacity, and studies show that the tenacity can be enhanced by adding certain materials like geological polymers (like fly ash) or fiber [6]. Modifier like fly ash (FA) can improve properties of magnesium oxychloride cement (MOC) because of the active SiO_2_ and Al_2_O_3_ content [7].

Fly ash (FA) is frequently used as a geological polymer material, with a highly reactive activity [8], large specific surface area, and extra absorbency [9]. Recent studies show that the addition of FA is positive to neutralize the extra water in a hydration reaction, and thus can reduce the moisture absorption and efflorescence of the magnesium adhesive [6]. Moreover, it can mitigate the heating speed in hydration reactions, and reduce the thermal expansion rate of a magnesium adhesive [10]. Under alkaline conditions, geological polymers (compounded by FA and magnesium adhesive) react and then generate hydrates with a dense micro-structure [11].

Zhang et al. [12] found FA can be added into concrete to improve its properties. It can increase the workability, reduce the porosity, and enhance the strength of the concrete when the FA content reaches 15%. Zhao researched the modification of MOC with the addition of FA and found that the addition of FA may prolong the curing time of MOC. When the addition of FA reached 20%, the strength and water resistance of MOC (28 days) were effectively enhanced [13]. Liu researched the influence on the physical mechanics of magnesium oxychloride cement concrete (MOCC) with the addition of FA and found that the addition of FA can increase the porosity, enlarge the volume of harmful holes, and also weaken the compressive strength of MOCC [14]. Zhang made fiberboard with urea-formaldehyde adhesive, which was modified by FA. When the addition of FA reached 10%, the 24h TS (thickness swelling) of the product decreased to 9.4%. When the addition of FA reached more than 10%, it could effectively weaken the waterproof or moisture proof property [15]. Niu studied particleboard made of wooden waste materials with inorganic adhesive and proved the mechanical properties and environmental conservation were improved [16]. Researchers studied particleboard made from groundnut shell and rice husk wastes with a blended formaldehyde adhesive, but its mechanical property was not good enough [17]. Huang studied the effects of FA, phosphoric acid, and nano-silica on the properties of MOC, found that MOC incorporating a combination of FA and H_3_PO_4_ demonstrated 1.5 times the strength retention of MOC with only FA added, after 28-days of immersion in water [18]. Recent research shows that using MOC as a replacement for Portland cement in the mixture of composite building materials where ordinary cement has some incompatibilities is meaningful for the environment and industry development [19].

The above-mentioned studies included MOC made with different adhesives and materials, as well as FA and its influence on MOC. Maybe a perfect way for manufacturing particleboard is still unknow. “Modify the magnesium oxychloride adhesive particleboard (MOPB) with the addition of FA” is a new idea in practice. In this study, modified MOPB with FA (MOPB-FA) was prepared to improve the mechanical properties of the composite. Additionally, the mechanism of modification and the bonding mechanism between the two materials were studied by analyzing the experimental data.

## 2. Materials and Methods

### 2.1. Materials

The materials required for this experiment were poplar shavings, an air-drying density of 0.386 g/cm^3^, and a water content of 10–12%. The shavings were irregular strips, divided into 50 mm × 2 mm × 1 mm and 5 mm × 2 mm × 1 mm according to the average size [20].

Magnesium oxychloride adhesive (MOA), the main chemical composition, refers to MgCl_2_ (including MgCl_2_ sulphate and Ca), MgO (including MgO, SiO_2_, CaO, etc.), and FA (including SiO_2_, Al_2_O_3_, Fe_2_O_3_, etc.). The main chemical composition ratio of raw materials is shown in Table 1.

### 2.2. Methods

Preparation of Magnesium Oxychloride Adhesive Particleboard (MOPB).

According to the raw material calculation method table (Table 2), high-precision electronic counting (ACS type, Zhejiang Jinhua City Wuyi Dahe Electronics Co., Ltd., Jinhua, China) was used to weigh a certain quality of raw materials (MgO, Shijiazhuang Tianyu Magnesium CO., Ltd., Shijiazhuang, China; MgCl_2_, Wuxi Yatai United Chemical Co. Ltd., Wuxi, China; and poplar shavings, Wen’an Wood-based panel factory, Langfang, China). First, prepare the brine, then burn the MgO powder into brine lightly, using a mixer for mixing mortars (JJ-5 type, Wuxi Construction Engineering Test Instrument Equipment Co., Ltd., Wuxi, China) to stir for 3 min in accordance with the “slow–fast–slow” way.

### 2.3. Preparation and Maintenance of FA Modified MOA (MOA-FA)

The proportion of FA added was calculated by the percentage of the weight of the modified MOA (Table 3). The modified adhesive was poured into the poplar shavings in stages and stirred well, and then the mixture was laid, pressed, and put into the cold press (CGYJ-100, Shijiazhuang Can High Frequency Machinery Co., Ltd., Shijiazhuang, China.). After 8 h of pressing, we placed the product in a constant temperature humidity box (DHS-225 type, Beijing North Lihui Test Instrument Equipment Co., Ltd., Beijing, China) within the maintenance to the corresponding age period. The preparation process for MOA-FA is shown in Figure 1.

### 2.4. Mechanical Property Test

The graduated cylinder was used to weigh the amount of MOA-FA, and then calculate its density. According to the requirements of national standard GB/T 14074-2006, the viscosity of MOA-FA was determined by a rotary viscometer (NDJ-5S, Shanghai Fangrui Instrument Co., Ltd., Shanghai, China). The Digital pH meter (PH-100, Shanghai Pingxuan Scientific Instrument Co., Ltd., Shanghai, China) was used to test the PH. We calculated the solid content of the specimen. The initial and final coagulation time of MOA-FA was measured using the Vika instrument (ISO-01, Cangzhou Xinding Experimentalr Instrument Co., Ltd., Cangzhou, China). The Contact Angle Measuring Meter (QCA 20, Audreno, Germany) measured the contact angle of MOA-FA (liquid) in contact with the surface of the wood chips (solid).

After the maintenance of the FA modified MOA particleboard (MOPB-FA), we recorded its weight after natural drying, then put it in the oven (101-1ES, Beijing Yongguang Co., Ltd., Beijing, China) to dry until the weight no longer changed, and then recorded the final weight. We calculated its moisture content. We calculated the actual density of MOPB-FA after measuring its weight using the electronic number display. In accordance with the provisions of the mechanics performance test method in GB/T 17657-1999 “Test methods of evaluating the properties of wood-based panels and surface decorated wood-based panels”, the modulus of rupture (MOR) and modulus of elasticity (MOE) were measured and recorded using the Magnum Test Machine (MMW-50 type, Jinan Naier Test Machine Co., Ltd., Jinan, China).

### 2.5. Detection and Characterization

The specimen was placed in an oven to become absolutely dry. Then, it was cut into 2 mm × 2 mm × 1 mm size. We used a field emission electron scanning microscope (JSM-7800F Prime, Japanese electrons) to observe its micromorphology and to select specific microscopic regions for energy dispersive system (EDS) scanning and analyzing. When the test piece was totally dry, we crushed it to 120 memes and prepared potassium chloride coated tablets for Fourier Transform Infrared Spectrometer (FTIR) analyzing, and we then crushed it to 60 memes and X-ray diffraction (XRD) analysis was carried out using an X-ray Powder diffractometer (Bruker D8, Brooke, Germany).

## 3. Results and Discussions

### 3.1. Analysis of the Performance of MOPB-FA

The previous study indicated that when the molar ratio of MgO/MgCl_2_ (M value) = 5, H_2_O/MgCl_2_ molar ratio (H value) = 15, MOA addition rate (R) = 70 wt%, cold pressing time (T) = 8 h, curing days (d) = 28 days, the mechanical properties of MOPB were the best. The study selected M = 5, H = 15, R= 70 wt%, FA addition amount of 20 wt% to prepare the FA modified adhesive. The performance analysis is shown in Table 4, after adding FA, adhesive seepage into the wood sped up, the solid content increased, initial viscosity increased, and the condensation time increased compared with before.

### 3.2. The Mechanical Properties of MOPB-FA

Figure 2 shows that when the amount of FA addition was 15–25 wt%, the mechanical strength of MOPB was better than before.

According to the requirements of GB/T 4897-2015 particleboard (Table 5), taking 6-13 mm thick particleboards as an example, with the addition of FA (15 wt%), the MOE value (4500 MPa) of MOPB was two times higher than the requirements of the standard P3-type bearing particleboard (2200 MPa). When the addition of FA was 20 wt%, MOR (16.32 MPa) was 1.48 times higher than the standard P3-type bearing particleboard requirements (11 MPa), FA modified MOPB with significant effects on the mechanical properties (Figure 2).

### 3.3. The Micro-Structure of MOA-FA

The micro-structure of the FA monomer was honeycomb-shaped with many unevenly sized pores (Figure 3a). The distribution of Mg, Cl, Al, and Si can be observed on the EDS diagram. With the addition of FA (15 wt%), the crystal structure shape of the MOA remained completely needle-shaped and rod-shaped. Moreover, it filled the pores of FA (Figure 3b). The micro-structure of MOA also became moderately loose, and the thickness of the hydrated layer (5.1.8-phase crystal structure) became moderately thinned, which could effectively enhance the elasticity of MOA.

The spherical Si molecule (Figure 4c) of FA can be observed in Figure 4, embedded in the MOA, and the whiskers of MOA crystal are adsorbed and wrapped in the Si molecule (Figure 4d). The Si molecule has a large specific surface area, FA contains SiO_2_ molecules, which have a large surface energy and reaction activity, so that the free molecules around absorbed on its surface, forming a gel system like the “skeleton”, effectively improved the toughness of the micro-structure system of adhesives. The modified adhesive had better elasticity when combined with wood. Compared with the value before modification, MOR (16.32 MPa) and MOE (4500 MPa) were significantly higher than before (13.11 MPa, 2600 MPa), with the addition of FA from 15 wt% to 25 wt% (Figure 2).

### 3.4. The Micro Structure of MOPB-FA

The micro morphology of MOPB-FA with poplar shavings glued is shown in Figure 5, when the amount of FA addition was 5 wt%, the MOA and wood were closer glued than before, with less pores and cracks. The section morphology showed that the wood was fully inserted into the adhesive rather than loosening or being pulled out (Figure 5a,b). At the same time, the particles of SiO_2_ in FA were small with a strong adsorption, so the modified MOA was more easily filled in the small pores of the wood, making the physical effect of “glue nail” of the adhesive more obvious (Figure 5c–e). As the addition of FA increased to 20 wt%, the chemical reactions between the Si molecules and MOA aggravated, and the whiskers wrapped the Si molecules (Figure 5f–g) little by little until they were completely encased. At this time, the monomers of Si molecules and adhesive crystals gradually polymerized into polymers, forming geological polymers and becoming a new gel system. Crystals with Si molecules of MOA were partially adsorbed on the micro surface of wood and partially interspersed in the micro pores of wood (Figure 5h–i), thus they were closer bounded to the wood pores and were more helpful for the bonding between the adhesive and the wood.

When the addition of FA was 35 wt% (Figure 5j–k), the whisker structure of MOA changed, it was no longer needle-bar-like and dense but became a dispersed structure that was flaky and flakelike. The reason may be that FA absorbed the moisture of MOA due to its strong adsorption, thus blocked and affected the hydrated reaction process, making the whisker structure of MOA incomplete. Studies on micro morphology showed that when the addition of fly ash was less than 25 wt%, it could effectively promote the generation of geological polymers, promote the generation of a Si gel system, and enhance the degree of bunding with wood. In this circumstance, it is much harder to separate the compounds.

### 3.5. The Chemical Composition of MOPB-FA

As can be seen in Figure 6, with the addition of FA, the characteristic peak of the -OH functional group of MOA shifted from 3319 cm^−1^ to 3411 cm^−1^ in the high band (Table 6), indicating that the associated reaction between the non-water -OH and -OH from the lignocellulose organism was reduced in the adhesive. At the same time, -OH from the lignocellulose organism was involved in the hydrated reaction, and reacted with Al, Cl, and other components in FA, and generated geological polymers like aluminosilicate. The -CH_2_- near the 1436 cm^−1^ band and the Si-O feature peak near the 1160 cm^−1^ band both increased, indicating that the silicate substances in FA decomposed and recomposed.

When the addition of FA was 20% (F_3_ group), the NH-stretching region of the -OH and Si-O bonds was relatively sharp, indicating that under the preparation of this addition, the Si element in FA mostly reacted in the chemical reaction with a strong reaction activity. The following details may explain the phenomenon. The Si-O bonds and Al-O bonds were excited by the alkaline conditions in MOA, thus the single-sided crystal molecular bonds gradually broke and depolymerized, forming a low-polymer silicon-oxygen tetrahedron and making a complexation reaction take place with the non-water -OH organic groups, free Cl, Mg, etc.

### 3.6. The Crystal Structure of MOPB-FA

Figure 7 shows that, in addition to the 5.1.8 phase, MgO and MgCO_3_ crystallization, there was also a C-Si-H crystal diffraction peak (C_13_H_36_Si_4_, hydrated calcium silicate) after adding FA, indicating that with the stimulation of alkalic Mg(OH)_2_, the Si-O and Al-O bonds in FA gradually broke and split, and chemical reactions occurred with -OH and active Cl ions in MOA, recombining to produce aluminosilicate, which gradually condensed into crystals with a polymer chain-like structure, and that eventually generated a hydrated calcium silicate C-Si-H (C_13_H_36_Si_4_) gel system [21].

With the different ratios of FA addition, the crystal structure of the XRD changed obviously. When the ratio was 15–20 wt%, the feature peak area and peak shape of the five-phase crystallization increased, and the five-phase crystal content increased, indicating that the hydrated reaction of MOA was more adequate. When the ratio was more than 20 wt%, the crystallization of MgCO_3_ disappeared, the H-value interacted with the ratio of FA addition, and when the ratio increased, the water required for the hydrated reaction was consumed, resulting in an inadequate hydrated reaction, so the crystal content reduced and the degree of crystallization decreased.

A new inter-wear network structure system was formed between MOA and FA and generated a C-Si-H gel system. The results of the mechanical test showed that the mechanical properties of FA were obviously enhanced, and the modified preparation of MOPB was effectively realized.

### 3.7. The Bonding Mechanism and Model of MOPB-FA

The mechanical properties of MOPB-FA were significantly improved (Table 7).

Considering from the view of physical combination, as shown in the Figure 8 model, MOA whiskers and the unique pore structure of FA formed a new three-dimensional skeleton structure in the microcosmic. MOA whiskers crossed through the holes of FA and FA was interspersed like “beads”. The larger Si molecules in FA were “set” in the whiskers, they “connected” with each other and formed a compact structure that was interspersed and interwoven. The structure made the crystal structure of MOA loose and continuous, and the thickness of the hydrated layer was moderately thinned, which effectively enhanced the elasticity of the MOA. At the same time, the structure could effectively play a buffering and protecting role when forced by external forces, so that the crystal structure system of MOA was not easy to be destroyed. Under the condition of external force, the structure was able to absorb stress by micro-deformation, thus it has the function of weakening and buffering external forces. Therefore, MOPB-FA has good elasticity, the material is not easy to break, and has the basis to be used as a wood-based composite adhesive.

Analyzing from the chemical reaction point, the basic structural units of FA are silicon-aluminum-oxygen chain (-Si-O-Al-O-), silicon-aluminum-silicon-oxygen chain (-Si-O-Al-O-Si-O), and silicon-aluminum silicon oxygen chain (-Si-O-Al-O-Si-O-Si-O-). AlO_4_ and SiO_4_ are tetrahedral structures and formed a three-dimensional network structure (Figure 9), with Si atoms in the middle and O atoms around. With a good reactivity, they can react with free hydroxyl and other active function groups of MOA.

Under the stimulation of alkaline, the process of “depolymerize-recompose” took place to form a new C-Si-H gel system. The system is a new enhanced structure of a modified geological polymer, the monomer structure is chain-like, with good denseness and enhanced toughness [22].

## 4. Conclusions

In this study, fly ash (FA) was added to modify MOPB. With analyses on its mechanical properties and the modification mechanism, some conclusions were obtained.

The use of FA effectively improved the mechanical strength of MOPB, when the ratio of FA addition was 15 wt%, MOPB-FA’s MOR value was 16.32 MPa, an increase of 24.5% more than before; the MOE value was 4595.51 MPa, an increase of 76.7% more than before; and the TS value was 0.35%, an increase of 85.2% more than before.

During the hydrated reaction, Si-O and Al-O ionic bonds in FA followed a process like “break-depolymerize-recompose”, recombined with -OH and active Cl ions in MOA to produce aluminosilicate. Moreover, they gradually polymerized into a hydrated calcium silicate C-Si-H (C_13_H_36_Si_4_) gel system with a polymer chain structure.

FA changed the internal hydrated layer structure of MOA, reconstituted the interwoven network structure interwoven with the five-phase crystal whisker, and effectively improved the mechanical elasticity of MOPB.

## Figures and Tables

**Figure 1 materials-15-02845-f001:**
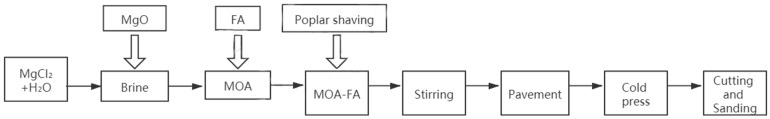
Production flow of MOA.

**Figure 2 materials-15-02845-f002:**
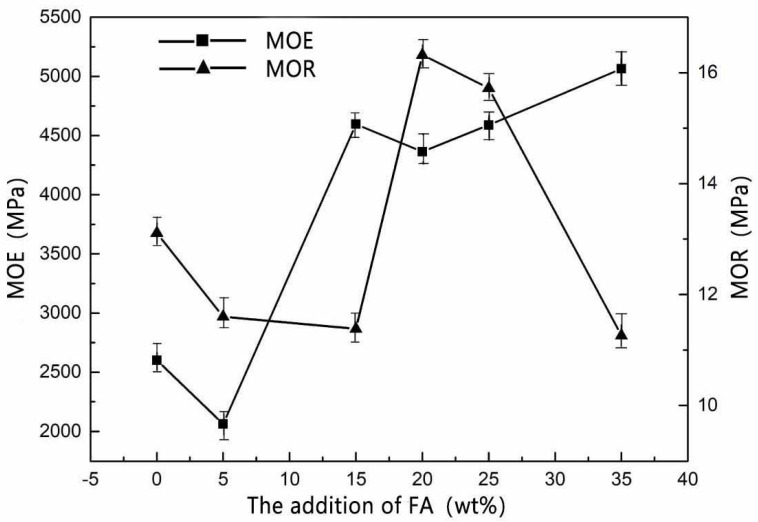
Mechanical strength of MOPB-FA (H = 10, M = 5).

**Figure 3 materials-15-02845-f003:**
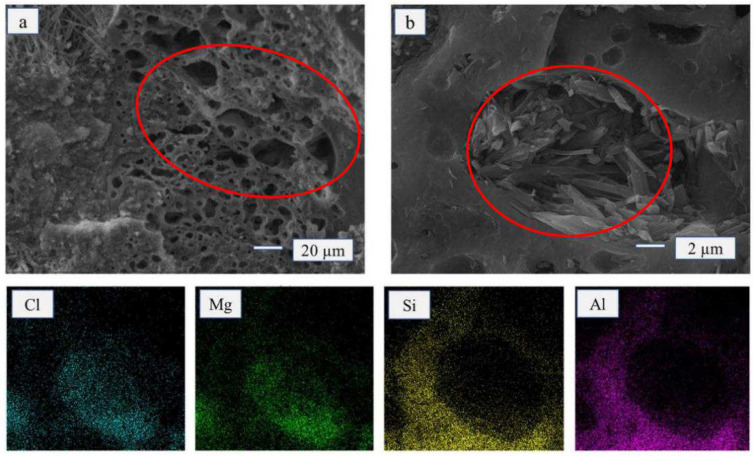
SEM and EDS of FA and MOA-FA. (**a**) FA; (**b**) MOPB-FA (15 wt%); (Cl, Mg, Si, and Al) EDS for the figure (**b**) area.

**Figure 4 materials-15-02845-f004:**
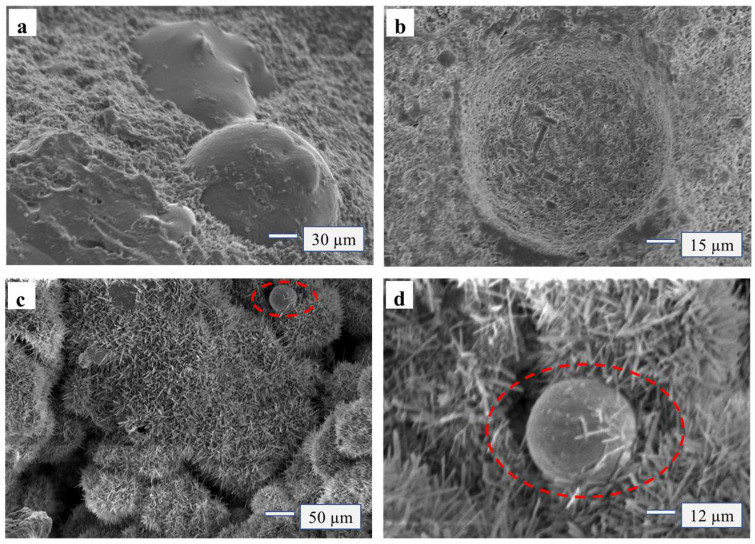
SEM of 5-phase crystallization and Si from FA. (**a, b**) EDS for different areas and magnifications; (**c**) the spherical Si molecule of FA; (**d**) the spherical Si molecule of FA.

**Figure 5 materials-15-02845-f005:**
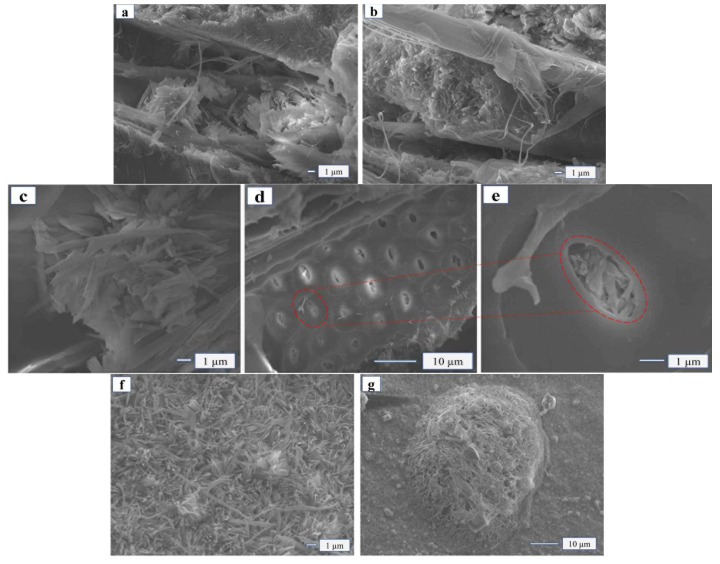
SEM of MOPB-FA with different FA additions. (**a**,**b**) 5 wt%; (**c**–**e**) 15 wt%; (**f**,**g**) 20 wt%; (**h**,**i**) 25 wt%; (**j**,**k**) 35 wt%.

**Figure 6 materials-15-02845-f006:**
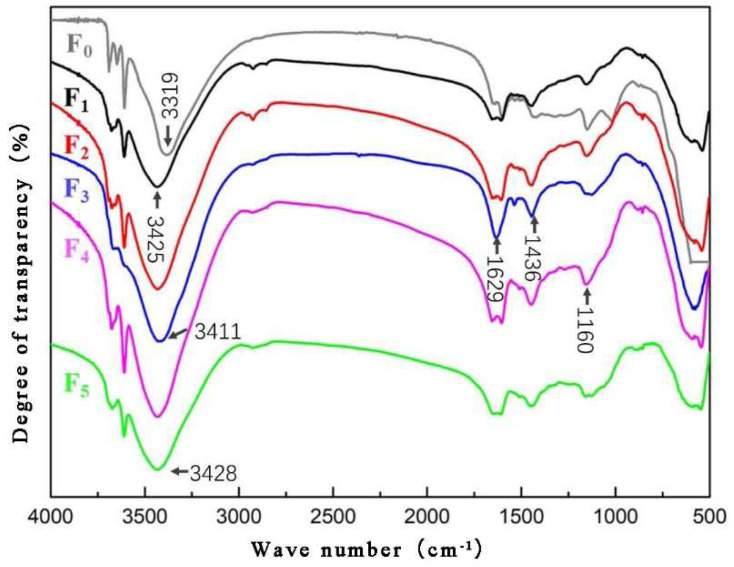
FTIR of MOPB modified by FA with different addition. FA addition: F_0_ = 0 wt%, F_1_ = 5 wt%, F_2_ = 15 wt%, F_3_ = 20 wt%, F_4_ = 25 wt%, F_5_ = 35 wt%.

**Figure 7 materials-15-02845-f007:**
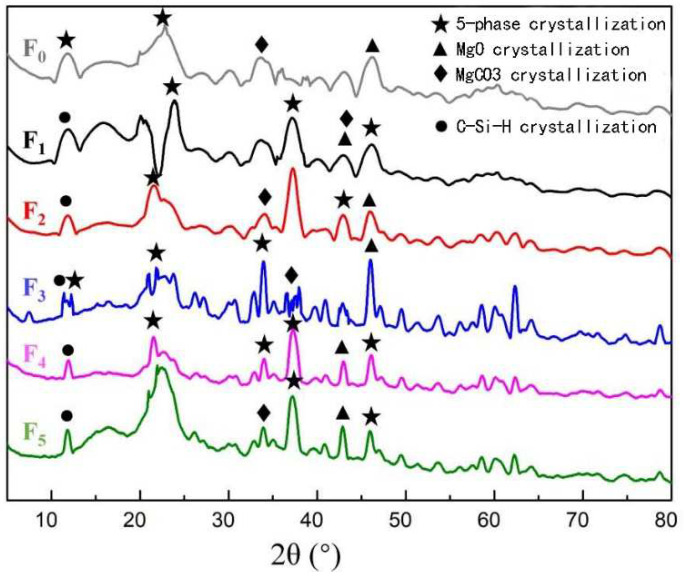
XRD of MOPB with different FA additions. FA addition: F_0_ = 0 wt%, F_1_ = 5 wt%, F_2_ = 15 wt%, F_3_ = 20 wt%, F_4_ = 25 wt%, F_5_ = 35 wt%.

**Figure 8 materials-15-02845-f008:**
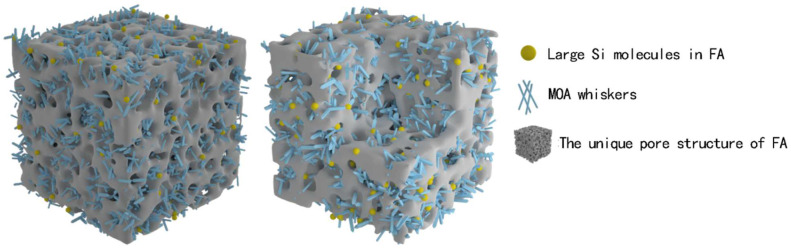
The model of MOPB-FA.

**Figure 9 materials-15-02845-f009:**
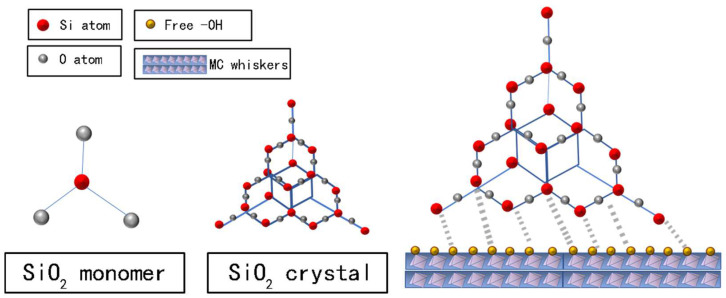
Model of chemical reaction of MOA and FA.

**Table 1 materials-15-02845-t001:** Ingredient list of main materials (%).

	Ingredients	MgCl_2_	MgO	SiO_2_	CaO	Al_2_O_3_	Fe_2_O_3_	Sulphate	Ca
Materials									
MgCl_2_		98.3	0	0	0	0	0	0.047	0.0048
MgO		0	86.26	6.04	1.12	0.48	0.34	0	0
FA		0	1.2	50.8	3.7	28.1	6.2		

**Table 2 materials-15-02845-t002:** Calculating methods of different raw materials.

Formulation Dosage	Calculation Method
The weight of MgO (g)	Weigh MgO according to the requirements
The amount of moles of MgO (M)	(The weight of MgO × 86.26%)/40.31
The amount of moles of MgCl_2_ (M)	MgCl_2_/MgO (molar ratio) × MgO (amount of moles)
The weight of MgCl_2_ (g)	(The amount of moles of MgCl_2_ × 95.21)/98.3%
The weight of H_2_O (g)	[MgCl_2_ (M) × (H_2_O/MgCl_2_ Molar ratio)] × 18
The weight of MOA (g)	The weight of MgO, MgCl_2_ and H_2_O.
The weight of MOBP (g)	The weight of MOBP/percentage of MOA (wt%)
The weight of poplar shavings (g)	The weight of MOBP × percentage of shavings (wt%)

**Table 3 materials-15-02845-t003:** Addition percent of FA.

Materials	Adding Percent (wt%)					
FA	0	5	15	20	25	35
MOA	100	95	85	80	75	65

**Table 4 materials-15-02845-t004:** Parameters of MOPB-FA (M = 5, H = 15, FA addition = 20 wt%).

Initial Viscosity (MPa·s)	Initial Set Time (min)	Final Set Time (min)	Initial Contact Angle (°)	Solid Content (%)	pH Value
22,615	465	510	120.9	85.2	7.05

**Table 5 materials-15-02845-t005:** Requirements of national standard GB/T 4897-2015 (thickness of 6–13 mm).

Property	P2-Type	P3-Type	P4-Type
	Furniture Particleboard	Bearing Particleboard	Overloaded Particleboard
MOR (MPa)	11	15	20
MOE (MPa)	1800	2200	3100
24 h TS (%)	≤8	≤19	≤16

**Table 6 materials-15-02845-t006:** Functional group of MOPB-FA.

Wave Number (cm^−1^)	Absorption Band Belonging and Explaining
3428	The NH-stretching region of -OH
3425	
3411	
3319	
1629	The NH-stretching region of -C=O
1436	The asymmetrical peak of -CH_2_-
1160	The NH-stretching region of Si-O bond
541	The NH-stretching region of Mg-O bond

**Table 7 materials-15-02845-t007:** Comparison of MOPB and MOPB-FA.

Parameter of Properties	MOPB	MOPB-FA	Increase Rate (%)	National Standard
MOR (MPa)	13.11	16.32	24.49	10.05
MOE (MPa)	2600.00	4595.51	76.75	1800.00

## Data Availability

The data presented in this study are available upon request from the corresponding author.

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
