# Peer review of "Study on the Properties and Mechanism of Magnesium Oxychloride Adhesive Particleboard Modified by Fly Ash"

_materials, 2022, doi:10.3390/ma15082845_

Round 1
Reviewer 1 Report
- Revise the introduction section thoroughly according to industrial applications. Add more literature and compare the findings.
- Highlight the novelty of research work.
- Re-write the abstract.
- Add error bar in the Fg. 2.
Overall, the paper is well written and presented. I recommended with minor revision.
Author Response
Dear Editors,
Thank you for your comments, they did do much help for this article. As for I am too naive in this area, so I’ve made some mistakes. Thank you again for your sincere help and patience.
During this revision, I found some mistakes caused by my carelessness. I am so sorry for my faults, I’ve revised and highlighted them. Thank you again for your sincere help.
My responses are as follows:
- Revise the introduction section thoroughly according to industrial applications. Add more literature and compare the findings.
R: Thank you for your comment, I’ve revised the introduction.
- Highlight the novelty of research work.
R: Thank you for your advice, here I mentioned it in the introduction.
- Re-write the abstract.
R: Thank you, I’ve revised it.
- Add error bar in the Fg. 2.
R: Sorry, it’s my ignorance. I’ve added. Thank you. All the error ranges are less than 5%.

Reviewer 2 Report
The paper has been improved. However, I still find the presentation in the manuscript is rather confusing. For example:
Line 119-126 – this part should not appear in the Results and Discussion section. Please move it to Methodology section. The authors should consider to revise the Methodology section extensively. The authors has done a lot of good works, as shown in the results and discussion section, however, this test was not mentioned in the Methodology section. Please revise it.
Abstract need improvement, only mechanical and physical properties was presented. But that’s not all the study that has been conducted in this study. Please revise.
The Introduction part also need slight improvement. Line 44-59, the authors listed out every previous study, please make it a good flow.
Table 4 - no comparison with other control adhesive?
Line 143 – what is static intensity?
Line 168-169 – can you please explain why addition of 5 wt% FA reduced MOR and MOE of the PB?
Line 195 – what is bunding degree?
Please revise the English language throughout the manuscript. In some part, I find it difficult to follow.
Author Response
Dear Editors,
Thank you for your comments, they did do much help for this article. As for I am too naive in this area, so I’ve made some mistakes. Thank you again for your sincere help and patience.
During this revision, I found some mistakes caused by my carelessness. I am so sorry for my faults, I’ve revised and highlighted them. Thank you again for your sincere help.
My responses are as follows:
- Line 119-126 – this part should not appear in the Results and Discussion section. Please move it to Methodology section. The authors should consider to revise the Methodology section extensively. The authors has done a lot of good works, as shown in the results and discussion section, however, this test was not mentioned in the Methodology section. Please revise it.
R: Thank you for your comment, I’ve moved it. Besides, I revised the Methodology section, briefly mentioned the test we did.
- Abstract need improvement, only mechanical and physical properties was presented. But that’s not all the study that has been conducted in this study. Please revise.
R: Thank you for your comment, I’ve revised it.
- The Introduction part also need slight improvement. Line 44-59, the authors listed out every previous study, please make it a good flow.
R: Sorry, I’ve changed it. Thank you.
- Table 4 - no comparison with other control adhesive?
R: Yes, maybe I could explain this. Here we only listed a comparatively better result. Others are shown in Figure 2. I know it’s not so proper to just show one set of data, but here we just mean to show the results, no comparation. I hope this would be OK. Thank you again.
- Line 143 – what is static intensity? It’s MOR.
R: Sorry, it’s a false. It means the modulus of rupture. I’ve changed it.
- Line 168-169 – can you please explain why addition of 5 wt% FA reduced MOR and MOE of the PB?
R: Yes, I could try to explain this for you. When the addition of FA was 5 wt%, the reaction between FA and MOPB was not enough to improve its MOR and MOE. On the contrary, it disturbed the reaction of polar shavings and MOA, that’s why the MOR and MOE decreased.
Besides, these articles may help me explain this question.
Effect of Fly Ash on Early Properties of Magnesium Oxychloride Cement
The impact of raw material ratio of magnesium oxychloride cement bamboo particleboard performance
My explanation may be similar to this in the article.
- Line 195 – what is bunding degree?
R: Here, it means the level/degree of bunding is much better than before.

Reviewer 3 Report
The topic of the manuscript "Study on the Properties and Mechanism of Magnesium Oxychloride Adhesive Particleboard modified by Fly Ash" is contemporary, intending to achieve the production of eco-friendly wood-based panels. Therefore, I think there will be interest in readers.
In the Abstract: Sentences, "The Magnesium Oxychloride Adhesive (MOA) is environmentally friendly for particleboard, fly ash (FA) may be helpful to improve the mechanical properties of the composite. Here, the study prepared modified Magnesium Oxychloride Adhesive Particleboard (MOPB) with FA 14 (MOPB-FA) to improve the mechanical properties of the composite and study the mechanism of 15 modification and bonding mechanism between the two materials" (lines 12-15) should be edited. In their current form, they are vague and poorly constructed.
In the Introduction: The analysis of previous research on the manuscript's topic should be significantly expanded—the authors' use of only twelve references ant it is highly insufficient in this rather extensive area.
In the Materials and Methods: The type and quantity of materials used (the experimental plan) need to be substantiated to a much greater extent. That is, to present similar research in the field and thus justify the study's parameters.
In the Results and Discussion: The study results are presented in detail, but there is no comparative analysis with similar studies. In my opinion, Table 5 is redundant, as these are well-known and easily accessible standards.
The Conclusions correspond to the achieved results. Here my recommendation is not to give them as a numbering but as a generalized conclusion of the results achieved in the study.
The References cited are appropriate but insufficient.
Author Response
Dear Editors,
Thank you for your comments, they did do much help for this article. As for I am too naive in this area, so I’ve made some mistakes. Thank you again for your sincere help and patience.
During this revision, I found some mistakes caused by my carelessness. I am so sorry for my faults, I’ve revised and highlighted them. Thank you again for your sincere help.
My responses are as follows:
The topic of the manuscript "Study on the Properties and Mechanism of Magnesium Oxychloride Adhesive Particleboard modified by Fly Ash" is contemporary, intending to achieve the production of eco-friendly wood-based panels. Therefore, I think there will be interest in readers.
- In the Abstract: Sentences, "The Magnesium Oxychloride Adhesive (MOA) is environmentally friendly for particleboard, fly ash (FA) may be helpful to improve the mechanical properties of the composite. Here, the study prepared modified Magnesium Oxychloride Adhesive Particleboard (MOPB) with FA 14 (MOPB-FA) to improve the mechanical properties of the composite and study the mechanism of 15 modification and bonding mechanism between the two materials" (lines 12-15) should be edited. In their current form, they are vague and poorly constructed.
R: Thank you for your comment, I’ve revised it.
- In the Introduction: The analysis of previous research on the manuscript's topic should be significantly expanded—the authors' use of only twelve references ant it is highly insufficient in this rather extensive area.
R: Thank you for your comment, I’ve revised it. I’ve read more and cited some. - In the Materials and Methods: The type and quantity of materials used (the experimental plan) need to be substantiated to a much greater extent. That is, to present similar research in the field and thus justify the study's parameters.
R: Thank you for your comment. I read some similar researches and designed our study. As for they could be found in the references in the Introduction part, and to shorten the article, I didn’t mention it in this part. - In the Results and Discussion: The study results are presented in detail, but there is no comparative analysis with similar studies. In my opinion, Table 5 is redundant, as these are well-known and easily accessible standards.
R: Thank you for your advice. For there may be some freshmen who lack knowledge about this, so I think table 5 could help. - The Conclusions correspond to the achieved results. Here my recommendation is not to give them as a numbering but as a generalized conclusion of the results achieved in the study.
R: Thank you for your advice, I’ve revised it.
- The References cited are appropriate but insufficient.

Round 2
Reviewer 2 Report
The paper has been improved accordingly. I think it is acceptable for publication.
Reviewer 3 Report
The esteemed authors have complied with or explained all my recommendations. In my opinion, this has contributed to increasing the clarity and hence the quality of the manuscript. That gives me a reason to recommend accepting the manuscript in its present form.